# Family and personal history of cancer in the *All of Us* research program for precision medicine

**Lauryn Keeler Bruce**[1]*, **Paulina Paul**[1], **Katherine K. Kim**[2], **Jihoon Kim**[1], **Theresa H. M. Keegan**[3], **Robert A. Hiatt**[4,5], **Lucila Ohno-Machado**[6], **On behalf of the All of Us Research Program Investigators**[7]¶

**1** Department of Biomedical Informatics, University of California (UC), San Diego, La Jolla, CA, United States of America, **2** Department of Public Health Sciences, UC Davis School of Medicine, Davis, CA, United States of America, **3** Division of Hematology and Oncology, Center for Oncology Hematology Outcomes Research and Training, UC Davis School of Medicine, Davis, CA, United States of America, **4** Department of Epidemiology & Biostatistics, UC San Francisco, San Francisco, CA, United States of America, **5** Helen Diller Family Comprehensive Cancer Center, UC San Francisco, San Francisco, CA, United States of America, **6** Section of Biomedical Informatics & Data Science, Yale University School of Medicine, New Haven, CT, United States of America, **7** All of Us Research Program, National Institutes of Health, Bethesda, Maryland, United States of America

¶ Membership of the All of Us Research Program Investigators are listed in the Acknowledgments.
* lbruce@ucsd.edu

**Data Availability Statement:** Data was obtained for this study via the All of Us Researcher Workbench. Access is free following an authorization and approval process that requires

## Abstract

The *All of Us* (*AoU*) Research Program is making available one of the largest and most diverse collections of health data in the US to researchers. Using the *All of Us* database, we evaluated family and personal histories of five common types of cancer in 89,453 individuals, comparing these data to 24,305 participants from the 2015 National Health Interview Survey (NHIS). Comparing datasets, we found similar family cancer history (33%) rates, but higher personal cancer history in the *AoU* dataset (9.2% in *AoU* vs. 5.11% in NHIS), Methodological (*e.g.* survey-versus telephone-based data collection) and demographic variability may explain these between-data differences, but more research is needed.

## Introduction

Family history plays an important role in the development and implementation of cancer screening strategies. As of 2020, breast, lung, prostate, and colorectal cancer (CRC) comprised the top four cancers in the United States, with breast, ovarian, and non-polyposis CRC (also known as Lynch syndrome), originating from inherited germline variants resulting in earlier disease onset. In women, breast, lung, and CRC account for ~50% of new cancer diagnoses, while prostate, lung, and CRCs account for ~43% of diagnoses in men. While only 5–10% of all cancers are thought to be hereditary, a family history of cancers caused by somatic variants, such as non-heritable CRC and breast cancer, portend an increased risk of developing family associated cancer [1]. Early screening based on family history is associated with increased survival rates [1, 2].

While US guidelines exist for collecting family history information for assessing risk and developing treatment plans, [3] few recent family history of cancer studies have been

registration, completion of ethics training, and acceptance of a data use agreement (https://allofus.nih.gov/). All NHIS data is also freely available from the National Health interview Survey website (https://www.cdc.gov/nchs/nhis/nhis_2015_data_release.htm).

**Funding:** L.K.B, K.K., and L.O.M are supported by a grant through the NIH Office of the Director (OT2OD026552). T.H.M.K is supported by a grant through the National Cancer Institute (P30CA093373). There was no additional external funding received for this study. The funders had no role in study design, data collection and analysis, decision to publish, or preparation of the manuscript.

**Competing interests:** The authors have declared that no competing interests exist.

published. In a 2006 study involving National Health Interview Study (NHIS) data from 27,000 individuals, [4] one in four individuals reported that a first-degree relative (FDR; i.e., parent, sibling, or child) had been diagnosed with one of the five cancers [5]. Additionally, a 2010 study found that 5–10% of 1,019 individuals surveyed had an FDR or second-degree relative with breast, colorectal, prostate, or lung cancer [6]. While informative, this study was limited in its sampling method, which involved randomly calling listed phone numbers.

The *All of Us* (*AoU*) Research program originated in 2018 with the goal of improving human health through precision medicine. As part of its mission, *AoU* gathered comprehensive patient data, including personal and family cancer history, along with numerous other items including biospecimens [7, 8]. Unique to *AoU* is its projected size of at least one million participants and oversampling from groups historically under-represented in biomedical research. Leveraging this *AoU* dataset, our study evaluated family and personal rates of breast, lung, prostate, colorectal, and ovarian cancers, and compared those estimates with the 2015 NHIS data, a database designed to be representative of the U.S. population [4]. Evaluating data from these complementary sources will improve understanding of US cancer prevalence rates and how family and personal histories relate.

## Materials and methods

This observational cross-sectional study involves use of *All of Us* v4 and 2015 NHIS database to calculate statistics for individuals self-reporting a family history of cancer and a subset also reporting a personal medical history of cancer. Individuals were then further categorized by demographics, education, annual household income, and insurance status. The same analyses were performed on both NHIS and *AoU* data

### Ethics statement

The work described here was proposed by *All of Us* Consortium members, reviewed and overseen by the program's Science Committee, and confirmed as meeting criteria for non-human subjects research by the *All of Us* Institutional Review Board. All research was carried out with the ethical standards set forth in the Helsinki Declaration of 1975.

### Project review and approval process

The work described here was proposed by *All of Us* Consortium members, reviewed and overseen by the program's Science Committee, and confirmed as meeting criteria for non-human subjects research by the *All of Us* Institutional Review Board. The initial release of data and tools used in this work was published recently.[9] Results reported are in compliance with the *All of Us* Data and Statistics Dissemination Policy disallowing disclosure of group counts under 20.

### *All of Us* system

This study was performed using the previously described *All of Us* Research Program within the *All of Us* Researcher Workbench, the cloud-based user interface where approved researchers can access and analyze de-identified data [9]. The *All of Us* database currently contains EHR, physical measurements and survey data. The *All of Us* dataset allowed for the categorization of into five race and ethnicity groups based on self-reported survey responses: Asian, Black, Hispanic, White, and Other. EHR, survey and physical measurement data were compiled by the *All of Us* Research Program, which has been previously described [10]. Participation in these surveys is optional for all responders and individual questions may be skipped:

the 'Family history' survey, which asks about first- and second-degree familial history of diseases; and the 'Personal Medical History' survey, which asks about each respondent's cancer status. Specifics of the surveys are available in the *Survey Explorer* within the Research Hub. This project was organized as a part of the *All of Us* consortium demonstration projects to help identify issues with the data or tools made available to researchers. This and other such projects have been published with corresponding code made available via the Researcher Workbench to promote transparency and reproducibility [11, 12].

### *All of Us* study cohort

The *All of Us* program provides surveys for individuals to complete as part of the program enrollment. This study employed the *AoU* database version 4 as of April 2021 that included data collected between May 30, 2017, and August 1, 2020, on a total of 383,808 individuals of which 314,994 completed 'The Basics' survey. This study uses the 'Basics', 'Family History', and 'Personal Medical history' survey data along with the demographic data from the *All of Us* Electronic Health Record (EHR) data. The Basics survey collects demographic information such as country of birth, self-identified race and ethnicity, biological sex, education level, and insurance status. The sub-cohort for this study comprises those who indicated an FDR with cancer on the 'Family History' survey. Specifically, only responses related to familial history for breast, prostate, colorectal, ovarian, or lung cancers were analyzed. Two additional surveys were employed in this study by a subset of 89,453 people: specifically, who, completed the 'Family History' survey, and 85,954 who completed the 'Personal Medical History' survey. Of those who responded to the 'Family History' and the 'Personal Medical History' questions, 82,142 individuals reported their personal cancer status.

### *All of Us* dataset preparation

The input for statistical analysis input was created in two steps, cohort building and dataset creation, using the *All of Us* research workbench. First, a cohort of those meeting the inclusion criteria was built. Only respondents who answered both 'The Basics' survey and the 'Family History' questions were included in this study. The survey questions were limited to the ones pertaining to the family history of FDR cancer status and cancer related options to the personal medical history question, ''Has a doctor or health care provider ever told you that you have any of the following?" The resulting cohort identified unique individuals. Second, a new dataset was created by adding the variables of interests as columns. The preliminary dataset created by a web-based *All of Us* research workbench tool was further processed to export an analysis-ready structured query language (SQL) code for subsequent statistical analyses in a Jupyter Notebook.

### *All of Us* statistical analysis

All AoU analyses were conducted using the Python 3.0 Jupyter Notebook (version 6.4.8, https://jupyter.org) in the *All of Us* researcher workbench. After loading the data using the exported structured query language (SQL) query, several intermediate data frames were created to clean, organize, normalize and convert the survey data from long to wide format with the *NumPy* (version 1.21.6, https://numpy.org), and *pandas* (version 1.3.5, https://pandas.pydata.org) python packages. The baseline characteristics table (Table 1) was constructed with the *tableone* package (version 0.7.12, https://pypi.org/project/tableone/) [13]. To ensure reproducible research, the cohorts, concept sets, datasets and the Python Jupyter notebooks are shared in the *All of Us* researcher workbench as a publicly available Featured Workspace Demonstration Project. Any table fields with counts representing fewer than 20 responders were masked to comply with *All of Us* policies. Two-sample proportion z-test was performed with

**Table 1. All of us (AoU) cohort and NHIS participant characteristics.**

| Demographic Variables | AoU v4 Family History | AoU v4 | NHIS Family History | NHIS |
|---|---|---|---|---|
| Sample | 89,453 | 383,808 | 24,305 | 76,261 |
| Age, mean (SD) | 54.9 (16.6) | 52.8 (17.4) | 53.9 (16.9) | 48.4 (17.4) |
| Age, median | 58 | 53 | 54 | 48 |
| Age groups, n (%) | - | - | - | - |
| 20–29 | 8,044 (9%) | 47,334 (12.3%) | 1,949 (8.1%) | 13,147 (17.4%) |
| 30–39 | 12,418 (13.9%) | 55,377 (14.4%) | 3,911 (16.2%) | 13,481 (17.8%) |
| 40–49 | 11,279 (12.6%) | 51,272 (13.4%) | 4,096 (16.9%) | 13,415 (17.7%) |
| 50–59 | 15,763 (17.6%) | 69,859 (18.2%) | 4,735 (19.6%) | 14,148 (18.7%) |
| 60–69 | 22,205 (24.8%) | 81,129 (21.1%) | 4,581 (18.9%) | 11,358 (15.0%) |
| 70–79 | 16,500 (18.4%) | 62,104 (16.2%) | 3,051 (12.6%) | 6,419 (8.5%) |
| 80+ | 3,024 (3.4%) | 14,707 (3.8%) | 1,879 (7.8%) | 3,659 (4.8%) |
| Biological Sex, n (%) | | | | |
| Female | 59,134 (66.6%) | 230,149 (60.0%) | 14,053 (57.8%) | 40,185 (52.7%) |
| Male | 29,713 (33.4%) | 148,819 (38.8%) | 10,252 (42.2%) | 36,076 (47.3%) |
| Race/Ethnicity, n (%) | | | | |
| Asian | 2,904 (3.3%) | 12,004 (3.2%) | 1,140 (4.7%) | 5,015 (6.6%) |
| Black | 6,701 (7.6%) | 76,817 (20.4%) | 3,057 (12.7%) | 9,491 (12.6%) |
| Hispanic | 6,921 (7.8%) | 66,312 (17.6%) | 3,903 (16.2%) | 14,670 (19.4%) |
| Non-Hispanic White | 69,198 (78.1%) | 208,670 (55.3%) | 15,517 (64.2%) | 44,534 (59.0%) |
| Another single population | 519 (0.6%) | 2,568 (0.7%) | 179 (0.7%) | 596 (0.8%) |
| More than one population | 1,575 (1.8%) | 6,836 (1.8%) | 365 (1.5%) | 1,146 (1.5%) |
| None of these | 728 (0.8%) | 3,935 (1.04%) | | - |
| Education, n (%) | | | | |
| Less than a high school degree or equivalent | 1,785 (2%) | 35,897 (9.6%) | 3,622 (15.0%) | 10,614 (14.2%) |
| Highest Grade: Twelve Or GED | 7,477 (8.4%) | 74,030 (19.7%) | 6,195 (25.7%) | 19,707 (26.4%) |
| Highest Grade: College One to Three | 20,318 (22.8%) | 101,228 (27%) | 4,471 (18.5%) | 14,036 (18.8%) |
| College graduate or advanced degree | 59,352 (66.7%) | 164,253 (43.8%) | 9,819 (40.7%) | 30,161 (40.5%) |
| Health Insurance, n (%) | | | | |
| Yes | 86,364 (97.4%) | 349,821 (93.6%) | 21,795 (90.3%) | 65,893 (88.1%) |
| No | 2,265 (2.6%) | 23,934 (6.4%) | 2,333 (9.7%) | 8,862 (11.9%) |
| Employment, n (%) | | | | |
| Employed for wages or self-employed | 47,185 (53.2%) | 162,380 (43.5%) | 13,563 (56.2%) | 46,764 (62.8%) |
| Not currently employed for wages | 41,481 (46.8%) | 210,590 (56.5%) | 10,568 (43.8%) | 27,691 (37.2%) |
| Household Income, n (%) | | | | |
| 0–25K | 4,464 (6.8%) | 56,694 (24.5%) | 4,243 (34.2%) | 13,648 (35.1%) |
| 25K–50K | 5,916 (9%) | 28,246 (12.2%) | 3,194 (25.7%)* | 10,507 (27.0%)* |
| 50K–75K | 12,966 (19.7%) | 40,473 (17.5%) | 2,831 (22.8%)** | 8,294 (21.3%)** |
| 75K–100K | 11,106 (16.9%) | 30,791 (13.3%) | 2,155 (17.3%)*** | 6,412 (16.5%)*** |
| 100K–150K | 14,654 (22.3%) | 36,938 (16%) | | |
| 150K–200K | 6,991 (10.6%) | 16,377 (7.1%) | | |
| > 200K | 9,691 (14.7%) | 21,858 (9.5%) | | |

GED: General Educational Development

*NHIS Reported for ranges: *25-45K, **45-75K, ***75K+

the proportion *ztest* function from the statsmodels (version 0.13.5, https://stats.models.org) package as documented on the package websites [14]. A p-value less than .05 was considered to indicate statistical significance.

### National health interview survey data preparation and analysis

A National Health Interview Survey (NHIS) was conducted by the National Center for Health Statistics in 2015 that included a Cancer Control Module that recorded an individual's family history of cancer [4]. The 2015 *Adult Cancer*, *Person*, and *Adult* NHIS survey databases were downloaded from the site https://www.cdc.gov/nchs/nhis/nhis_2015_data_release.htm. Respondent characteristics were extracted from the *Person* database, family history of cancer for each individual was extracted from the *Adult Cancer* database, and personal history of cancer was extracted from the *Adult* table. All fields were recategorized to replicate *All of Us* data and all NHIS analyses were conducted using Python 3.0 Jupyter Notebook on a local system.

## Results

The cohort in the *All of Us* program who completed the family health history (FH) survey does not display the same level of diversity as seen in the full AoU dataset v4. The *All of Us* FH cohort, as shown in Table 1, has a higher proportion of Whites, females, and those with higher education levels than those in the entire *All of Us* dataset. The demographic composition for the NHIS survey population and the FH subset are comparable for education and income while race/ethnicity, biological sex, and age are slightly skewed White, Female, and older, respectively. In comparison to the NHIS FH demographics, the *All of Us* FH cohort is older, has higher levels of education, health insurance rates, employment, and income. A contingency table chi-squared test indicated a significant association ($p << 0.001$, 95% confidence interval) between counts for each demographics category (e.g., age bin, sex, *etc*.) and the cohort (AoU or NHIS), thus indicating the two cohorts are different in demographics composition. The *All of Us* FH cohort is also underrepresented for racially Asian and Black groups and ethnically Hispanic individuals when compared to NHIS FH participants.

In the *All of Us* FH cohort, 32.75% (29,300) of responders reported having a FDR with a history of at least one of the five highlighted cancers, similar to NHIS FH participants (n = 7,967, 32.78%). In the *All of Us* cohort, FH of only a single cancer type was reported for 25.4% of responders, while 6.2% reported two cancer types and 1.2% reported three or more cancer types. The prevalence of family history of cancer in the *All of Us* cohort was 13.7% for breast, 9.18% for prostate, 8.71% for lung, 7.38% for CRC, and 2.44% for ovarian (Table 2). Analyses were also conducted according to the following responder demographic categories: age group, race/ethnicity, sex at birth, income, and highest education completed. The counts, ratios, and rankings for the *AoU* cohort for each demographic and cancer type are shown in FH S1 Table. In all five types of cancer, the percentage of respondents with family history of cancer was found to be higher in older age groups than in younger groups with the highest proportions generally reported between 60–80 years of age. For Asian responders, breast and colorectal cancer were the highest reported FDR cancer types, with overall 15.19% reporting at least one FDR with any of the five cancer types. Self-reported Black, Hispanic, non-Hispanic White, and Other individuals showed the highest prevalence of breast and prostate cancers among FDR relatives. About 25% of Black, 18% of Hispanic, 31% of White, and 26% of Other individuals reported family history of any of the five cancers. Results by sex-at-birth, limited to female and male, showed very similar prevalence between sexes with the highest percentage difference for prostate cancer at only 1.22%. Some differences were seen based on income level, for example, the rate of prostate cancer is almost 3x higher for individuals with an income >200k. The highest family history of cancer prevalence by education was for those who completed some or all of college.

Similar analyses were performed for the *NHIS* FH participants. The prevalence of family history of cancer in *NHIS* FH participants was 12.47% for breast, 7.33% for prostate, 9.71% for

**Table 2. Respondents' family history of cancer (FH) and personal history (PH) of cancer, by cancer type and study.**

| Cancer Type | All of Us | | | | | | NHIS | | | | | |
|---|---|---|---|---|---|---|---|---|---|---|---|---|
| | FH * | | PH ** | | Both ** | | FH *** | | PH **** | | Both **** | |
| | n | % | n | % | n | % | n | % | n | % | n | % |
| Breast | 12255 | 13.7 | 3865 | 4.7 | 1029 | 1.3 | 3030 | 12.5 | 569 | 2.3 | 163 | 0.7 |
| Colorectal | 6599 | 7.4 | 597 | 0.7 | 94 | 0.1 | 1817 | 7.5 | 185 | 0.8 | 40 | 0.2 |
| Lung | 7789 | 8.7 | 377 | 0.5 | 81 | 0.1 | 2361 | 9.7 | 93 | 0.4 | 21 | 0.1 |
| Ovarian | 2185 | 2.4 | 286 | 0.4 | 21 | 0.03 | 624 | 2.6 | 72 | 0.3 | 8 | 0.03 |
| Prostate | 8213 | 9.2 | 2048 | 2.5 | 495 | 0.6 | 1782 | 7.3 | 359 | 1.5 | 96 | 0.4 |
| Total Unique | 29300 | 32.8 | 7173 | 8.7 | 1720 | 2.1 | 7967 | 32.8 | 1241 | 5.1 | 671 | 2.8 |

* The denominator is the number of respondents to the *All of Us* family medical history survey (n = 89,453)

** The denominator is the number of respondents of both *All of Us* family and personal history surveys (n = 82,142)

*** The denominator is the number of respondents of both NHIS family history surveys (n = 24,305)

**** The denominator is the number of respondents of both NHIS family and personal history surveys (n = 24,288)

PH = Personal History Cancer

FH = Family History Cancer

lung, 7.48% for CRC, and 2.57% for ovarian (Table 2). We compared the two cohorts by ranking the prevalence ratios calculated in each study for 5 different demographic categories: sex at birth (n = 2), race/ethnicity (n = 5), age group (n = 7), income (n = 4), and education level (n = 4). Both cohorts reported breast cancer as the highest ranked cancer in all demographic categories and ovarian cancer as the lowest ranked in all but the *NHIS FH* 20–29 age group, where colorectal was the lowest ranked (S2 Table). *NHIS* FH participants showed lung cancer as the second most prevalent in all demographics, except for four categories (i.e., those who self-identified as Hispanic, aged 20–29, had an income greater than 75,000, or had an education level of college or higher). In contrast, the *All of Us* FH cohort ranked prostate cancer as the second highest in 14 of 22 categories and ranked lung cancer second only in 7 of the 22 categories, notably in older age categories (n = 2), lowest income (n = 2) and education levels (n = 3).

The highest reported prevalence was for breast cancer, for both the *All of Us* and *NHIS*, at 13.7% and 12.5%, respectively (Table 2). In the *All of Us* cohort, of those responders who reported a FH of breast cancer, 11,291 responders (12.6%) reported having only a single FDR who has been diagnosed with breast cancer. For all cancer types, almost all had only one relative (92%) with cancer, a small percentage (7.7%) had two, and only 85 had three or more (0.3%) (Table 3). The proportion of respondents who reported two family members with cancer was highest for breast and lung cancer, at 7.5% and 7.3%, respectively. Three or more first-degree relatives were only reported for breast, colorectal, and lung cancer, at 0.33%, 0.26%, and 0.32%, respectively. Similar proportions were found in the NHIS dataset.

In the *All of Us* FH cohort, among those who also provided their personal history of cancer, 9.2% had a PH of cancer and 2.1% reported having both FH and PH of the same cancer (Table 2). Breast and prostate cancers were the most and second most prevalent both in FH and PH, whereas history of lung cancer was third for FH but fourth for PH. Ovarian cancer was ranked lowest for both. The results from the *NHIS* survey showed a lower prevalence of PH of cancer overall, but a higher prevalence of having both a PH and FH of cancer (Table 2). The prevalence of PH of cancer was ranked the same for both datasets, with breast at the top and ovarian the bottom. Analyses of PH of cancer rates were also conducted for the same five demographic categories as FH producing the counts, ratios, and rankings for both cohorts for each demographic and cancer type (S3 and S4 Tables).

**Table 3. Proportion of respondents with one or at least two first degree relatives (FDR) by type of cancer and study.**

| Cancer Type of FDR | All of Us | | | | NHIS | | | |
|---|---|---|---|---|---|---|---|---|
| | One FDR | | At least two FDR | | One FDR | | At least two FDR | |
| | n | % | n | % | n | % | n | % |
| Breast | 11291 | 92.1 | 924 | 7.5 | 2827 | 93.3 | 202 | 6.7 |
| Colorectal | 6161 | 93.4 | 421 | 6.4 | 1691 | 93.1 | 121 | 6.7 |
| Lung | 7191 | 92.3 | 571 | 7.3 | 2211 | 93.7 | 139 | 5.9 |
| Ovarian | 2124 | 97.2 | 61 | 2.8 | 608 | 97.4 | 16 | 2.6 |
| Prostate | 7743 | 94.3 | 467 | 5.7 | 1687 | 94.7 | 94 | 5.3 |
| Total | 34510 | 93.2 | 2444 | 6.6 | 9024 | 93.9 | 572 | 6 |
| Unique Total | 27855 | 92.04 | 2324 | 7.7 | 6515 | 81.8 | 1290 | 16.2 |

FDR = First Degree Relative

We also examined the relationships between PH and FH of cancer. Of those respondents that reported a personal breast cancer history, both *AoU* and NHIS reported that >34% of responders had one FDR with breast cancer, > 11% had two FDRs, and >2.2% had three or more FDRs with breast cancer (Table 4). For the *All of Us* cohort, the conditional probability of reporting a PH of cancer given at least 1 family history of cancer was 32% (8,620/27,007). This probability given no FH was 20.4% (11,242/55,135) (Table 5). The same conditional probabilities in *NHIS* were lower at 16.3% and 7.4%, respectively. Table 6 shows these conditional probabilities by type of cancer. For example, in the *All of Us* cohort, 10.13% (1,197/11,814) of respondents reported a PH and FH of breast cancer, while 14.38% (11,814/ 82,152) reported FH of breast cancer. A PH of any type of cancer (not limited to the highlighted five) and a FH of one of the five cancers was reported by 32.88% (8,620/27,007) of respondents.

To understand the conditional probabilities of PH of cancer and FH of cancer for race/ethnicity and sex-at-birth subsets, the probability of both personal history and family history given personal history of cancer (%PH & FH given PH) and of both personal and family history given family history of cancer (%PH&FH given FH) were calculated (S5 Table). For the probability given a PH, the *All of Us* cohort shows breast cancer as the highest probability in all but the Asian subgroup, though notably, Asians are underrepresented in this *All of Us* subset. Given a FH of cancer, all race/ethnicity groups show breast cancer with the highest probability, colorectal cancer is the next highest-ranked cancer for White and Hispanic subgroups, while

**Table 4. The proportion of respondents who reported having a personal history (PH) of a specific cancer and reported having 1, 2, or 3 or more FDRs with that same cancer type by study.**

| PH Cancer Type | All of Us | | | | | | NHIS | | | | | |
|---|---|---|---|---|---|---|---|---|---|---|---|---|
| | One FDR | | Two FDR | | > = 3 FDR | | One FDR | | Two FDR | | > = 3 FDR | |
| | n | % | n | % | n | % | n | % | n | % | n | % |
| Breast | 1347 | 34.9 | 428 | 11.1 | 77 | 2 | 223 | 39.2 | 76 | 13.4 | 15 | 2.6 |
| Colorectal | 190 | 31.8 | 60 | 10.1 | <20 | 2.5 | 79 | 42.7 | 22 | 11.9 | 3 | 1.6 |
| Lung | 127 | 33.7 | 28 | 7.4 | <20 | 4 | 30 | 32.3 | 16 | 17.2 | 1 | 1.1 |
| Ovarian | 84 | 29.3 | 26 | 9.1 | <20 | 4.1 | 26 | 36.1 | 8 | 11.1 | 2 | 2.8 |
| Prostate | 706 | 34.4 | 214 | 10.4 | 61 | 3.1 | 135 | 37.6 | 54 | 15 | 10 | 2.8 |
| Total | 2454 | 72.1 | 756 | 22.2 | 192 | 5.6 | 493 | 70.4 | 176 | 25.1 | 31 | 4.4 |
| Unique Total | 2454 | 34.2 | 756 | 10.5 | 192 | 2.7 | 460 | 38.1 | 153 | 12.7 | 31 | 2.6 |

FDR = First Degree Relative

**Table 5. Counts of respondents with and without personal history of cancer by family history status.**

| | All of Us | | | NHIS | | |
|---|---|---|---|---|---|---|
| | ≥ 1 Family member with Cancer (%) | No Family member Cancer (%) | Total n (%) | ≥ 1 Family member with Cancer (%) | No Family member Cancer (%) | Total n (%) |
| **Responder with Any Cancer** | 8,620 | 11,242 | 19862 (24.2) | 1,930 | 915 | 2845 (11.7) |
| **Responder without Any Cancer** | 18,387 | 43,893 | 62280 (75.8) | 9,928 | 11,515 | 21443 (88.3) |
| **Total** | 27007 (32.9) | 55,135 (67.1) | 82,152 | 11858 (48.8) | 12,430 (51.2) | 24,288 |

ovarian cancer was the next highest in the Black subgroup. For analyses by sex-at-birth, the NHIS subgroup probabilities given a FH of cancer were consistently lower than the *AoU* cohort; however, probabilities given a PH were similar in *AoU* and NHIS.

## Discussion

Family cancer history is a recognized risk factor in many cancer types [5, 6] and is used to inform clinical recommendations regarding screening and referral to a specialty cancer genetics clinic [15, 16]. Here we report rates for both family and personal history of cancer in the 2021 *AoU* cohort compared with the 2015 NHIS study. We found higher rates of PH of cancer in *AoU* than in NHIS, 9.2% and 5.11% respectively, but similar prevalence of FH of cancer overall in *AoU* and NHIS (33%). Notably, the conditional probability of having personal cancer, given at least one family member has had one of the five types of cancer, was almost double in *All of Us* compared with NHIS and more than double if the individual did not report a family history of cancer. Like in previous studies, we found that the prevalence of FH and PH of cancer varies by age, race/ethnicity, income and education, and sex [5, 6].

In *AoU*, 32.8% of responders reported a FH of one or more cancers, which is almost identical to the prevalence found in *NHIS participants*. Of the individuals who also provided their PH of cancer, 9.2% of the *All of Us* cohort reported a PH and 2.2% reported both FH and PH of cancer, while *NHIS* participants conveyed a lower personal cancer rate of 5.11% and similar rate of having both a FH and PH of cancer, 2.76%. Reported cancer rates were highest for

**Table 6. Personal history (PH) of one of five cancers and family history (FH) for the same cancer.**

| | All of Us | | | | NHIS | | | |
|---|---|---|---|---|---|---|---|---|
| Cancer Type | PH & FH of this cancer type | FH of this cancer type | FH specified cancer type within those who completed PH | Rate of this cancer type in respondents, given family history of this cancer type* | PH & FH of this cancer type | FH of this cancer type | FH specified cancer type within those who completed PH | Rate of this cancer type in respondents, given family history of this cancer type** |
| Breast | 1,197 | 11,814 | 14.40% | 10.10% | 163 | 3,030 | 12.50% | 5.40% |
| Colorectal | 138 | 6,364 | 7.80% | 2.20% | 40 | 1,816 | 7.50% | 2.20% |
| Lung | 106 | 7,495 | 9.10% | 1.40% | 21 | 2,360 | 9.70% | 0.90% |
| Ovarian | 29 | 2,098 | 2.60% | 1.40% | 8 | 624 | 2.60% | 1.30% |
| Prostate | 577 | 7,968 | 9.70% | 7.20% | 96 | 1,782 | 7.30% | 5.40% |
| PH Any Cancer and FH 1 of the 5 cancers | 8,620 | 27,007 | 32.90% | 31.90% | 671 | 7,965 | 32.80% | 8.40% |

PH = Personal History Cancer; FH = Family History Cancer

* Denominator = 82,152

** Denominator = 24,288

older individuals, aligning with the general assumption that family members are more likely to have cancer the older they are. In both studies, FH cancer rates were similar between females and males, and an upward trend with increased income was observed for all cancers, except for ovarian cancer. For all race/ethnicity groups, breast cancer showed the highest proportion, while either Asian or Hispanic populations had the lowest rate for each cancer type.

Population sampling was conducted differently for *AoU* and NHIS and may have impacted subgroup cancer prevalence. While participation in both studies was voluntary, joining the *AoU* study required additional commitments, such as submission of biosamples and physical measurements, including blood for whole genome sequencing, and provision of access to electronic health records. Several studies have shown that a significant portion of the population is not yet ready to share data from electronic health records with all researchers [17, 18]. Additionally, enrollment of *AoU* participants occurred in participating clinics with a recruitment goal of including underrepresented minorities. It is also possible that individuals in the *AoU* Research Program have increased interest in certain health conditions, particularly those that may have a hereditary component [19]. Thus, it is no surprise that the recalling of FH or PH of cancer seems to be higher in the *AoU* cohort.

One goal of the *All of Us* program is to recruit individuals in demographic groups that are systemically underrepresented in medical research and reference databases, including racial and ethnic minority groups and those in lower income brackets, with disability status, or without access to health services [10]. While the full *All of Us* population has a more diverse composition than the US population, with 20% of responders identifying as Black and 17% identifying as Hispanic, respondents to the optional FH survey are less diverse, with only 7.5% Black and 7.7% Hispanic. The overall demographics of this survey sub-cohort showed a higher average age, proportion of women, rate of those who received college degrees or higher, ratio with health insurance, and a higher proportion of responders who identify as non-Hispanic White. These trends are not reflected in the 2015 NHIS FH cohort, which maintain similar proportions of individuals in the race/ethnicity, education, and income demographics when compared to its full cohort. For *All of Us*, categories that show higher than average prevalence, such as health insurance and completion of higher levels of education, may be partially due to enrollment of many individuals through hospitals and clinics in health care systems, which is the main method employed.

Family history is an important risk factor in the five common cancers considered. In CRC, previous studies have found that a higher proportion of patients with CRC diagnosed at 50 years of age or younger have a family history of the disease when compared to individuals diagnosed after this age [20, 21]. In addition, the risk for CRC is highest for individuals with two or more FDR or other relatives with earlier onset disease [15, 22]. For breast cancer, 5–10% are thought to be hereditary [23] and family history of breast cancer has been associated with a greater than 60% increase in risk [24]. Prostate cancer has been found to be one of the most heritable cancers, with a FDR history of prostate cancer associated with a 68% increase in total risk. Twin studies reported that about 57% of risk of prostate cancer can be explained by germline genetic determinants [25]. In addition, a family history of breast cancer is associated with a 21% increase in total risk of developing prostate cancer. For lung cancer, relative risk was reported to almost double when comparing individuals with one or more FDR to those with three or more FDRs diagnosed [26]. Lastly, for ovarian cancer, family history can increase risk for individuals 3- to 7-fold, with higher prevalence for those with more than one relative or FDR diagnosis at younger [27, 28].

The *All of Us* program's goals have created a unique dataset that is expanding with each new version. Uniquely, the consistent design of the database and user interface tools will allow for the analyses performed here to be repeated easily in future releases using the code shared

on the Researcher Workbench. The additional level of voluntary participation in the family and personal history surveys has possibly introduced biases and reduced the overall diversity, however, with continued enrollment of a diverse population, this database will be an important cancer resource available researchers both in the United States and across the world.

## Conclusions

We provide rates of family history and personal history of five cancer types from two large and diverse publicly available datasets, the *NHIS* 2015 survey and *All of Us* v4 released April 2021. In both the *All of Us* and *NHIS* data sets, 33% of responders have at least one first-degree family member who has been diagnosed with cancer. Personal history of cancer was higher in the *AoU* group than the NHIS, 9.2% versus 5.1%, possibly due to optional survey participation via the *AoU* website. Conditional probabilities of reporting a personal history of cancer given at least 1 family history of cancer was 32% and 16.3% for *All of Us* and *NHIS*, respectively. The *All of Us* code methods and dataset are open source and available to any researcher agreeing to the terms stated by the *All of Us* program. The methods within this study may be used to provide updated statistics as the *All of Us* program grows to meet its goal of enrolling one million individuals.

## Supporting information

**S1 Table. *All of Us* family history of cancer by demographic categories rates, counts, and ranking.**
(DOCX)

**S2 Table. NHIS family history of cancer by demographic categories rates, counts, and ranking.**
(DOCX)

**S3 Table. *All of Us* personal history of cancer by demographic categories rates, counts, and ranking.**
(DOCX)

**S4 Table. NHIS personal history of cancer by demographic categories rates, counts, and ranking.**
(DOCX)

**S5 Table. Conditional probability of personal and familial history of cancer given a personal history of cancer with regards to sex at birth or race and ethnicity.**
(DOCX)

**S1 File. Acknowledgement list of principal investigators.**
(DOCX)

## Acknowledgments

The *All of Us* Research Program is supported by the National Institutes of Health, Office of the Director: Regional Medical Centers: 1 OT2 OD026549; 1 OT2 OD026554; 1 OT2 OD026557; 1 OT2 OD026556; 1 OT2 OD026550; 1 OT2 OD 026552; 1 OT2 OD026553; 1 OT2 OD026548; 1 OT2 OD026551; 1 OT2 OD026555; IAA #: AOD 16037; Federally Qualified Health Centers: HHSN 263201600085U; Data and Research Center: 5 U2C OD023196; Biobank: 1 U24 OD023121; The Participant Center: U24 OD023176; Participant Technology Systems Center: 1 U24 OD023163; Communications and Engagement: 3 OT2 OD023205; 3 OT2

OD023206; and Community Partners: 1 OT2 OD025277; 3 OT2 OD025315; 1 OT2 OD025337; 1 OT2 OD025276. This work relies on the program organized and executed by the All of Us Research Program Investigators (S1 File). In addition, the *All of Us* Research Program would not be possible without the partnership of its participants.

## Author Contributions

**Conceptualization:** Katherine K. Kim, Theresa H. M. Keegan, Robert A. Hiatt, Lucila Ohno-Machado.

**Data curation:** Lauryn Keeler Bruce, Paulina Paul, Jihoon Kim.

**Formal analysis:** Lauryn Keeler Bruce, Paulina Paul.

**Funding acquisition:** Katherine K. Kim, Lucila Ohno-Machado.

**Methodology:** Lauryn Keeler Bruce, Paulina Paul, Katherine K. Kim, Lucila Ohno-Machado.

**Project administration:** Lauryn Keeler Bruce.

**Supervision:** Katherine K. Kim, Lucila Ohno-Machado.

**Visualization:** Lauryn Keeler Bruce, Paulina Paul.

**Writing – original draft:** Lauryn Keeler Bruce.

**Writing – review & editing:** Lauryn Keeler Bruce, Katherine K. Kim, Jihoon Kim, Theresa H. M. Keegan, Robert A. Hiatt, Lucila Ohno-Machado.

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
