## [Decision Letter · Decision Letter 0]

28 Apr 2023

PONE-D-23-04133Family and Personal History of Cancer in the All of Us Research Program for Precision MedicinePLOS ONE

Dear Dr. Bruce,

Thank you for submitting your manuscript to PLOS ONE. After careful consideration, we feel that it has merit but does not fully meet PLOS ONE’s publication criteria as it currently stands. Therefore, we invite you to submit a revised version of the manuscript that addresses the points raised during the review process.

Please follow closely all suggestions made by the reviewers and either accept and amend the manuscript accordingly or share why you decide not to follow the guidance. 

We look forward to receiving your revised manuscript.

Kind regards,

Michal Rosen-Zvi

Academic Editor

PLOS ONE

Journal Requirements:

"L.K.B, K.K., and L.O.M are supported by a grant through the NIH Directors office (OT2OD026552). T.H.M.K is supported by a grant through the National Cancer Institute (P30CA093373)."

"L.K.B, K.K., and L.O.M are supported by a grant through the NIH Directors office (OT2OD026552). T.H.M.K is supported by a grant through the National Cancer Institute (P30CA093373)."

"No authors have competing interests"

6. One of the noted authors is a group or consortium All of Us Consortium.. In addition to naming the author group, please list the individual authors and affiliations within this group in the acknowledgments section of your manuscript. Please also indicate clearly a lead author for this group along with a contact email address.

7. Your ethics statement should only appear in the Methods section of your manuscript. If your ethics statement is written in any section besides the Methods, please move it to the Methods section and delete it from any other section. Please ensure that your ethics statement is included in your manuscript, as the ethics statement entered into the online submission form will not be published alongside your manuscript. 

Additional Editor Comments:

This paper is well written and contributes to the understanding of cancer prevalence rates in US and associations of family and personal histories with these rates. It is based on analysis of All of Us data collection and compared with participants from the 2015 National Health Interview Survey (NHIS). It is a comprehensive intriguing analysis. The paper can be further improved as suggested by both reviewers. See their comments.

Reviewers' comments:

Reviewer's Responses to Questions

**Comments to the Author**

1. Is the manuscript technically sound, and do the data support the conclusions?

Reviewer #1: Yes

Reviewer #2: Yes

2. Has the statistical analysis been performed appropriately and rigorously? 

Reviewer #1: Yes

Reviewer #2: Yes

3. Have the authors made all data underlying the findings in their manuscript fully available?

Reviewer #1: Yes

Reviewer #2: Yes

4. Is the manuscript presented in an intelligible fashion and written in standard English?

Reviewer #1: Yes

Reviewer #2: Yes

5. Review Comments to the Author

Reviewer #1: The manuscript describes important findings for rates of certain types of cancers such as breast, colorectal, lung, prostate etc. from AoU cohort and compares it with NHIS cohort from self-reported family history (FH) and personal history (PH) survey questionnaires. Overall, the paper is well written, but is rather long and can be described more concisely in Results. It’s not clear what the main findings are – “we found similar family cancer history (33%) rates, but higher personal cancer history, with rates of personal cancer of nearly double in AoU as compared to the NHIS (9.2% vs. 5.11%)”. If it’s the latter, it will help the reader if they could describe results more concretely and point to the relevant tables.

Other comments -

Page 11-

a. 314,994 individuals who completed ‘The Basics’ survey � this number is larger in Table 1 (383,808) if its AoU v4

b. Table 2, foot note lists denominator as 89,458 where as the text lists as 89,453 from AoU with FH

c. Similarly, for AoU with FH + PH lists 89,142 vs. 89,152 in the text

Table 6, Page 22 – need N besides percentages for specific cancer types – in other words need to know how many in each bin for calculating conditional probability. This is the table that shows double rates?

Interestingly, colorectal cancer conditional probability remains same (2.20%) between AoU and NHIS cohorts in Table 6, while Breast cancer is doubled for AoU (10.10% vs 5.40%). Why such discrepancy? It may be also good to discuss, if it’s an artifact of data collection/survey methods?

Reviewer #2: This interesting manuscript compares rates of personal cancer history and family cancer history in the All of Us database and in NHIS. The paper is well written and technically describes surveys on large enough datasets (383,808 in All of Us, and 76,261 in NHIS), as well as in sub-cohorts of these datasets. The authors performed several summary statistics of the data, which can be seen in the tables in the main text and supplemental material. The code and datasets are open and available for any researcher who registers and completes the ethics training of All of Us.

Specific points to be addressed:

1.Materials and methods, Page 10: “with the hypothesis that prevalence statistics and trends would be comparable despite the differences in population composition”. If the study is hypothesis-driven, it would be ideal to state the hypothesis in the introduction. Materials and methods should state only well-described methods that can effectively answer the questions you are addressing.

2.Race and ethnicity categories were collected in the All of Us Research Program. Is it possible to also include the source of the classifications used (eg, self-report or selection, investigator observed, database, electronic health record, survey)?

3.Section “All of Us Study Cohort”: For those who are not familiar with the AoU database, what exactly does “the Basics” survey include? Which information does it contain? Please describe in one sentence or two.

4.Page 12, section “All of Us Dataset Preparation”, 1st sentence: word input was repeated twice.

5.Statistical Analysis, page 13: When stating the statistical methods used to analyze the data (two-sample proportion z-test), also state the p-value used for significance (was it .05? Then add “A p-value less than .05 was considered to indicate statistical significance.”

6.When mentioning statistical softwares/packages you used, also state version and manufacturer's name. Eg: We used numpy (version X.X.X, https://numpy.org) and pandas (version X.X.X., https://pandas.pydata.org). Same for jupyter notebook, and other libraries you used. This allows others to reproduce the study.

7.Table 1: Spell out the abbreviation “FH” in the table legend. The abbreviations E1, E2, E3 were defined but they do not appear in the table.

8.Maybe there is a typo in the first paragraph of Results: “the same level of diversity in seen”. Did you mean "as seen"?

9.Provide statistical significance (p-values) when comparing variables (race/ethnicity, age, biological sex) between cohorts. This is important to verify that your statistical analyses are performed to a high technical standard and are described in sufficient detail.

10.Under Results: “The demographic composition for the NHIS survey population and the FH subset are comparable for education and income while race and ethnicity are slightly skewed White, Female, and older.” Please rephrase this sentence. Perhaps “ […] while race/ethnicity and biological sex are slightly skewed towards White and female, respectively”

11.Discussion: “Like in previous studies, we found that the prevalence of FH and PH of cancer varies by age, race/ethnicity, income and education, and sex.” – can you cite which studies found these main observations?

12.This sentence is also missing a citation: “In CRC, previous studies have found that a higher proportion of patients with CRC diagnosed at 50 years of age or younger have a family history of the disease when compared to individuals diagnosed after this age.”

13.What were the limitations of your study? Was it the selection biases of AoU or NHIS recruitment that perhaps makes these two populations not comparable? Please state clearly in the discussion the limitations and biases inherent in your study.

6. PLOS authors have the option to publish the peer review history of their article (what does this mean?). If published, this will include your full peer review and any attached files.

Reviewer #1: No

Reviewer #2: **Yes: **Vesna Barros

---

## [Author Response · Author response to Decision Letter 0]

23 Jun 2023

Reviewer #1

The manuscript describes important findings for rates of certain types of cancers such as breast, colorectal, lung, prostate etc. from AoU cohort and compares it with NHIS cohort from self-reported family history (FH) and personal history (PH) survey questionnaires. Overall, the paper is well written, but is rather long and can be described more concisely in Results. It’s not clear what the main findings are – “we found similar family cancer history (33%) rates, but higher personal cancer history, with rates of personal cancer of nearly double in AoU as compared to the NHIS (9.2% vs. 5.11%)”. If it’s the latter, it will help the reader if they could describe results more concretely and point to the relevant tables.

Response: Thank you for the comment, we have simplified this statement in the Abstract.

1. Page 11- 314,994 individuals who completed ‘The Basics’ survey this number is larger in Table 1 (383,808) if its AoU v4

Response: Thank you for highlighting this discrepancy. The All of Us database v4 does have 383,808 participants, and of those, 314,994 individuals completed the Basics survey. We clarified this in the “All of Us Study Cohort” Section. 

2. Table 2, footnote lists denominator as 89,458 whereas the text lists as 89,453 from AoU with FH

Response: Thank you for catching this discrepancy, 89,453 was the total n used in the code, and the typo has been fixed in the Table 2 footnote. 

3. Similarly, for AoU with FH + PH lists 89,142 vs. 89,152 in the text

Response: Thank you for reporting this discrepancy. The correct value was 82152 participants that completed the personal cancer and history of cancer survey, thus the value in the Table 2 footnote was correct. The text has been changed to reflect the correct value. 

4. Table 6, Page 22 – need N besides percentages for specific cancer types – in other words need to know how many in each bin for calculating conditional probability. This is the table that shows double rates?

Response: Thank you for requesting clarification. Table 6 contains all n values used for calculating the conditional probability. For example, the probability of FH of Breast cancer given the participant completed the PH survey is equal to 11814/82152 = 14.4%. For the rate of breast cancer for a participant given that they also have a family history (PH|FH) is equal to 1197 / 11814 = 10.1%.

5. Interestingly, colorectal cancer conditional probability remains same (2.20%) between AoU and NHIS cohorts in Table 6, while Breast cancer is doubled for AoU (10.10% vs 5.40%). Why such a discrepancy? It may be also good to discuss, if it’s an artifact of data collection/survey methods?

Response: Thank you for highlighting this interesting difference. We do not believe the differences can be attributed to data collection methods. 

Reviewer #2: 

This interesting manuscript compares rates of personal cancer history and family cancer history in the All of Us database and in NHIS. The paper is well written and technically describes surveys on large enough datasets (383,808 in All of Us, and 76,261 in NHIS), as well as in sub-cohorts of these datasets. The authors performed several summary statistics of the data, which can be seen in the tables in the main text and supplemental material. The code and datasets are open and available for any researcher who registers and completes the ethics training of All of Us. Specific points to be addressed:

1. Materials and methods, Page 10: “with the hypothesis that prevalence statistics and trends would be comparable despite the differences in population composition”. If the study is hypothesis-driven, it would be ideal to state the hypothesis in the introduction. Materials and methods should state only well-described methods that can effectively answer the questions you are addressing.

Response: Thank you for this comment. We have removed the hypothesis statement from the Materials and methods section.

2. Race and ethnicity categories were collected in the All of Us Research Program. Is it possible to also include the source of the classifications used (eg, self-report or selection, investigator observed, database, electronic health record, survey)?

Response: Thank you for requesting this clarification. All race and ethnicity data were self-reported in the ‘The Basics’ survey. We have now stated this in the methods section.

3. Section “All of Us Study Cohort”: For those who are not familiar with the AoU database, what exactly does “the Basics” survey include? Which information does it contain? Please describe in one sentence or two.

Response: Thank you for highlighting the need for describing “The Basics” survey. Additional information has been added to the “All of Us Study Cohort” section. 

4. Page 12, section “All of Us Dataset Preparation”, 1st sentence: word input was repeated twice.

Response: Thank you for identifying that typographical error, this has now been resolved. 

5. Statistical Analysis, page 13: When stating the statistical methods used to analyze the data (two-sample proportion z-test), also state the p-value used for significance (was it .05? Then add “A p-value less than .05 was considered to indicate statistical significance.”

Response: Thank you for requesting this clarification, the statement suggested has been added to the statistical analysis section as a p-value of 0.5 is considered as statically significant for this study. 

6. When mentioning statistical softwares/packages you used, also state version and manufacturer's name. Eg: We used numpy (version X.X.X, https://numpy.org) and pandas (version X.X.X., https://pandas.pydata.org). Same for jupyter notebook, and other libraries you used. This allows others to reproduce the study.

Response: Thank you for the suggestion, all versions and websites have now been added for the packages listed in the statistical analysis section. 

7.Table 1: Spell out the abbreviation “FH” in the table legend. The abbreviations E1, E2, E3 were defined but they do not appear in the table.

Response: Thank you for the suggestion. ‘FH’ is no longer abbreviated and the E1-E3 definitions have been removed. 

8. Maybe there is a typo in the first paragraph of Results: “the same level of diversity in seen”. Did you mean "as seen"?

Response: Thank you for identifying this typo, we have replaced ‘in’ with ‘as’

9. Provide statistical significance (p-values) when comparing variables (race/ethnicity, age, biological sex) between cohorts. This is important to verify that your statistical analyses are performed to a high technical standard and are described in sufficient detail.

Response: We are not able to compare cancer rates directly between the two cohorts as these populations are significantly different by a chi-squared test (p << 0.01) and it is known that the All of Us cohort was not designed to be representative of the population. We instead compare conditional probabilities of personal history given family history (Table 5, S5). 

10. Under Results: “The demographic composition for the NHIS survey population and the FH subset are comparable for education and income while race and ethnicity are slightly skewed White, Female, and older.” Please rephrase this sentence. Perhaps “ […] while race/ethnicity and biological sex are slightly skewed towards White and female, respectively”

Response: Thank you for the comment, the sentence has been rephrased as suggested.

11. Discussion: “Like in previous studies, we found that the prevalence of FH and PH of cancer varies by age, race/ethnicity, income and education, and sex.” – can you cite which studies found these main observations?

Response: Thank you for highlighting the need to add citations, the citations for the two studies have now been added to this sentence. 

12. This sentence is also missing a citation: “In CRC, previous studies have found that a higher proportion of patients with CRC diagnosed at 50 years of age or younger have a family history of the disease when compared to individuals diagnosed after this age.”

Response: Thank you for asking for citations, two additional citations have been added to support this statement. 

13. What were the limitations of your study? Was it the selection biases of AoU or NHIS recruitment that perhaps makes these two populations not comparable? Please state clearly in the discussion the limitations and biases inherent in your study.

Response: Thank you for the question. The main limitation in comparing the two populations (AoU and NHIS) is the method of recruitment. We highlighted in the 3rd and 4th paragraphs of the discussion that the AoU study required additional commitments (biosample submission, sharing of personal health records, etc.), recruitment occurred in participating clinics, and in general, the overall cohort is designed to include a higher proportion of underrepresented minorities while the family and personal history of cancer data is collected via surveys that are not required.

---

## [Editor Report · Decision Letter 1]

29 Jun 2023

Family and Personal History of Cancer in the All of Us Research Program for Precision Medicine

PONE-D-23-04133R1

Dear Dr. Bruce,

We’re pleased to inform you that your manuscript has been judged scientifically suitable for publication and will be formally accepted for publication once it meets all outstanding technical requirements.

Kind regards,

Michal Rosen-Zvi

Academic Editor

PLOS ONE

---

## [Editor Report · Acceptance letter]

7 Jul 2023

PONE-D-23-04133R1 

Family and Personal History of Cancer in the *All of Us* Research Program for Precision Medicine 

Dear Dr. Keeler Bruce:

I'm pleased to inform you that your manuscript has been deemed suitable for publication in PLOS ONE. Congratulations! Your manuscript is now with our production department. 

Kind regards, 

on behalf of

Prof. Michal Rosen-Zvi 

Academic Editor

PLOS ONE